# A Modular and Semantic Approach to Personalised Adaptive Learning: WASPEC 2.0

Ufuoma Chima Apoki [1],* and Gloria Cerasela Crisan [1,2]

1    Faculty of Computer Science, Alexandru Ioan Cuza University, 700506 Iasi, Romania; ceraselacrisan@ub.ro
2    Faculty of Sciences, Vasile Alecsandri University of Bacau, 600115 Bacau, Romania
*    Correspondence: ufuomaapoki@gmail.com; Tel.: +40-726-989-772

**Abstract:** The ubiquity of smart devices and intelligent technologies embedded in e-learning settings fuels the drive to tackle the grand challenge of personalised adaptive learning. Personalised adaptive learning, which combines the core concepts of personalised learning and adaptive learning, attempts to take individual needs and features into account for personal development through adaptive adjustment. Personalised adaptive learning is supported at its heart by efficient real-time monitoring of the learning process and robust managerial capabilities, which are driven by data, as well as human intuition. The absence of reusable personalised content and logic is one of the key limitations of systems that adopt personalised learning. This is mostly due to the fact that business logic is frequently entangled with the system's primary functionality. As a result, such systems are unable to interact with other systems that do not adhere to identical design standards. The application of modular frameworks and the semantic web has the potential to be leading technologies that foster reusable personalised content and systems that can efficiently share information. WASPEC, a modular framework for personalised adaptive learning, is evaluated in this paper. An improved architecture, WASPEC 2.0, ensuring more flexibility is also presented in the concluding sections.

**Keywords:** learning management systems; personalised learning; adaptive learning; ontologies; semantic web; pedagogical agents; personalisation parameters

## 1. Introduction

Initially, e-learning systems were solely intended to provide students with material asynchronously; however, with the emergence of personal computers and the internet, they evolved into more interactive systems. Institutions such as the Open University, which largely focused on distance learning, took advantage of the Internet's capabilities to provide course material by e-mail rather than post or mail [1]. E-learning platforms of today are effective at tracking learning activities and managing learning processes, both asynchronously and synchronously.

With the growing number of organisations and institutions utilising these systems, one important shortcoming is the inability to accommodate diversity in learner needs and preferences, which may influence learning. Personalised adaptive learning is a concept that seeks to address that. Personalised adaptive learning is not a new notion; it is the outcome of integrating the goals of personalised learning with the pedagogical and technological opportunities that new ICT (Information and Communications Technology) tools bring.

Personalised adaptive learning is a combination of technologies that support Intelligent Tutoring Systems (ITS) and Adaptive Hypermedia Systems (AHS), a group of systems referred to by Brusilovsky and Peylo [2] as Adaptive and Intelligent Web-based Educational Systems. The major goal of these systems is to support personalised and adaptive learning by utilising domain information, learner knowledge, and teaching methodologies. AHS leverage information from the learner model to give personalised instruction specific to each learner, whereas ITS provide the same level of personalisation for all learners by incorporating artificial intelligence technology [2,3].

There are two broad techniques that academics and educators typically use to build and implement the architecture of personalised learning. The first approach would be to design the educational system from the ground up, with all of its various components and rules for personalisation. The second strategy entails complementing existing educational systems that do not provide default personalisation options.

The first approach has some drawbacks, including the difficulties of sharing personalisation logic and content owing to compatibility issues, as well as inflexibility in terms of extensibility and application to multiple domains. This is due to the fact that these systems deliver personalisation in a specific domain, and the personalisation strategies are built in the learning content or instruction. The learner model is also largely dependent on how the system's adaptive engine is developed. AHS, for example, uses various learner models to tailor the content and links of educational hypermedia spaces to the learners [2]. Even if the different models in the architecture are separated, the personalisation strategies are frequently interwoven in the domain model, learner model, or the system's business logic. As a result, the business logic for personalisation is so entangled with the system's basic operations that modification is impossible without interfering with other components unrelated to personalisation. To use various pedagogical models or personalisation strategies in a different domain, the learner model, content model, and system's business logic would typically need to be re-authored.

The second approach suggests a modular design in which personalisation is provided to e-learning systems as an additional functionality. The business logic for personalisation is embedded in the architecture of a domain-independent system using this method. The engine that supports personalisation is completely independent from the essential functionalities of an e-learning system, such as storing learning content and basic learner information. This modular method is used in the design of WASPEC, a Weighted Agent System for Personalised Curriculum [4], which is implemented by extending Moodle [5]. The modular approach makes use of semantic web technologies, the implementation of learning objects, and pedagogical agents to deliver multi-parameter personalisation of learning resources to various courses on an e-learning platform.

The WASPEC platform was first presented to domain experts and researchers who were familiar with personalised learning, and the results were published in a previous study [4]. This report focuses on describing the evaluation of the WASPEC platform conducted with students of Petra Christian Academy, Ughelli. Ontologies and semantic rules are employed for knowledge representation in the current WASPEC platform implementation. Personalisation is also accomplished through activities (facilitated by web service communication) on a web platform built with Laravel [6], a PHP open-source framework. To encourage increased modularity among personalisation components, we propose an updated WASPEC design that will be completely implemented as a MOODLE plugin.

The paper is organised as follows in this context: Section 2 explores the general architecture of personalised adaptive learning; Section 3 describes the brief architecture of the WASPEC platform; Section 4 presents and analyses the results of the evaluation with students; Section 5 discusses the architecture of WASPEC 2.0 in detail; Section 6 presents related work, as well as how the WASPEC platform compares to similar designs in the literature; and Section 7 concludes the paper and discusses future directions.

## 2. The Framework of Personalised Adaptive Learning

According to Office of Educational Technology [7], "*Personalized learning refers to instruction in which the pace of learning and the instructional approach are optimized for the needs of each learner. Learning objectives, instructional approaches, and instructional content (and its sequencing) may all vary based on learner needs. In addition, learning activities are meaningful and relevant to learners, driven by their interests, and often self-initiated.*" Personalised learning, which incorporates the core values of personal needs, personal development, and individual differences [8] into curriculum, assessment, and pedagogical approaches, can be implemented in and out of the classroom, using technology in scenarios such as blended or

flipped learning or exclusively in e-learning. Other concepts used interchangeably in the literature to refer to personalised learning in the context of e-learning, both technically and pedagogically, include [8–10]:

- *Adaptive learning* refers to learning scenarios where learners get learning content that is personalised to their needs and preferences via the use of ICT technologies.
- *Individualised learning* refers to when students study at their speed and according to their own needs.
- *Competency-based learning* presents a scenario in which learners demonstrate competence as they advance toward various learning objectives. Competencies are a coherent application of information, skills, beliefs, resources, experiences, and tools that may be utilised while addressing a problem or (successfully) engaging in a task.
- When learners are offered alternative learning approaches based on their preferences, this is referred to as *differentiated learning*.

The similarity between these many types of personalised learning yields the following common views: the intersecting correlations of the various forms, the inclusion of other forms in personalised learning, and the inclusion of other forms of personalised learning in adaptive learning. While the final viewpoint is fairly frequent (and occasionally improperly applied), a closer look at the definitions and values indicates that these strategies can be applied differently or integrated, as shown in Figure 1. Despite considerable differences, the definitions share the following characteristics [10]: 1. Each learner's learning speed is tailored to their needs; 2. learning goals, techniques, material, and tools are tailored to their needs; 3. the learner's interests, choices, and context are crucial to the learning process; and 4. technology may help students learn more successfully.

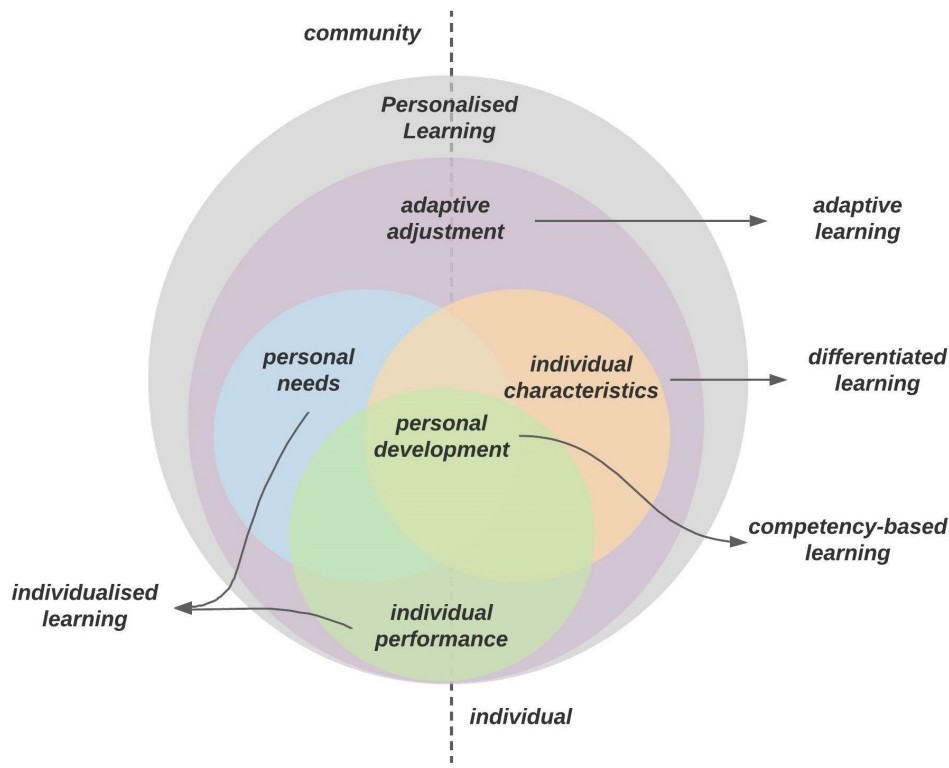

**Figure 1.** Different Forms of Personalised Learning.

Other more in-depth definitions of adaptive learning include the following:

- *"Adaptive learning strategies create a student experience that is modified based on a students performance and engagement with the course materials. At its heart is an approach to instruction that relies on technology and data about student performance to adjust and respond with*

*content and methodologies that develop a pathway to the student's mastery of a particular learning objective*" [11].

- "*Adaptive learning refers to the technologies monitoring student progress, using data to modify instruction at any time*" [12].

Adaptive learning, according to these definitions, is data-driven, replicates some sort of intelligence, and may occur in real-time, in addition to the use of technology, which is the major emphasis. This sets it apart from typical e-learning tools, which are solely used to augment learning. Individual differences, performance, and needs are all important components in adaptive learning, as they are in personalised learning. While personalised learning may be applied in a variety of methods, adaptive adjustment is the essential aspect of adaptive learning [8]. Thus, personalised adaptive learning is an educational approach that focuses on the personal growth of learners by integrating technology and sound pedagogies, taking into consideration learners' unique preferences, needs, and performance, and is supported by both adaptive adjustment and human feedback.

In learning scenarios, personalised adaptive learning integrates data and human intuition in decision-making to answer the issues of "*what to learn*", "*how to learn*", and "tex-tithow well learnt". The link between decision-making and the key values of personalised adaptive learning is depicted in Figure 2.

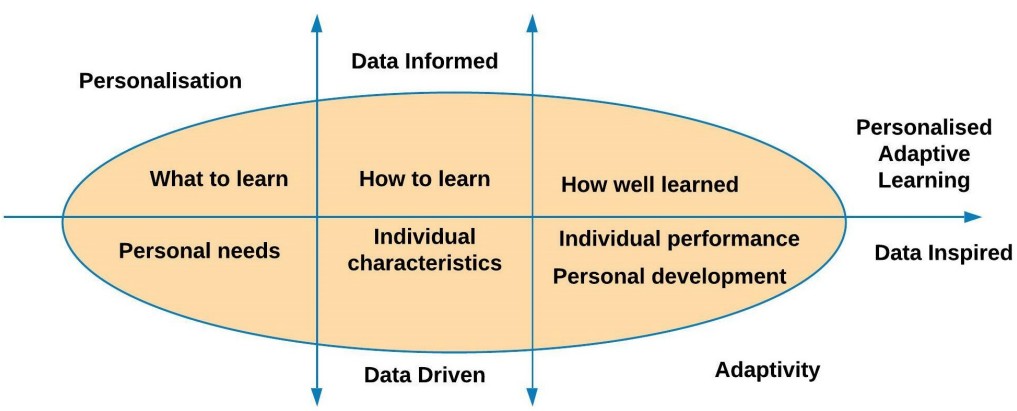

**Figure 2.** The Framework of Personalised Adaptive Learning.

Data has the final say in data-driven approaches. Data from learners and interactions with the system are utilised to automate the learning process. Along with pedagogical research and significant human insight, data is a key component of the data-informed approach. Decisions are made after taking into account all of these variables. A data-inspired approach examines trends and incorporates them into the consideration of creative ideas to further strengthen the learning process via multiple rounds of the learning process and data insights.

## 3. The Architecture of WASPEC

Figure 3 depicts the WASPEC architecture, which consists of the following primary components: an Learning Management System (LMS), a Service Framework (SF), a Semantic Framework (SmF), and a Multi-Agent system (MAS).

The platform is accessed by the learners via the LMS, which is Moodle for this implementation. The learners' interactions with the platform, as well as exercises and questionnaires, are used to determine their preferences and needs. This data is saved in the learner model. Personalisation is implemented in the SF (designed with Laravel). On the SF, the course instructor specifies personalisation criteria and metadata specification items. These are used for semantic annotation of existing courses and content on the LMS.

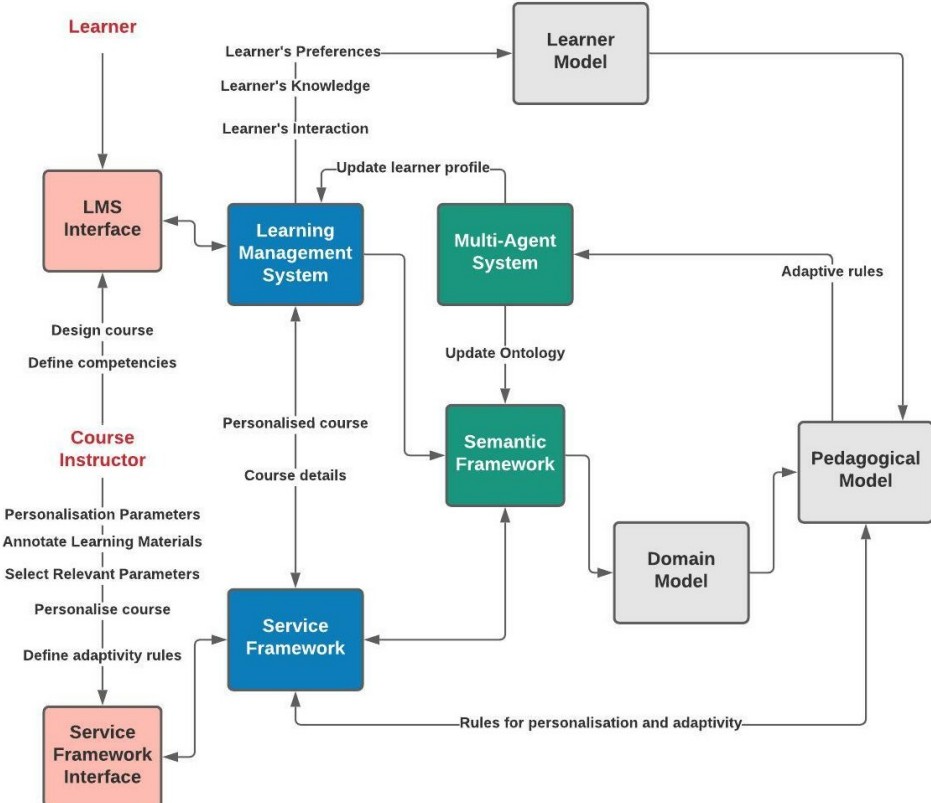

**Figure 3.** The Architecture of WASPEC.

Web services are used by the SF to communicate with the LMS. When application programs need to communicate with one another, web services provide effective interoperability solutions. A service, in this sense, is a remote function that is independent of the implementation and can be run when called [13]. The usage of web services offers the flexibility to "only call the services when needed, allowing for efficient integration of personalisation technologies" [4]. The web services enable numerous LMS functions, including semantic annotation of LMS courses, educational materials, and learners.

The SF and LMS data repositories are specified as graph databases in the SmF and represented in EOMPP (E-learning Ontology for Multi-Parameter Personalisation), allowing the selection and combining of relevant personalisation criteria for each course [4]. The classes of EOMPP, designed for the implementation of the WASPEC platform, are based on the elements of the IEEE Learning Object Metadata (LOM) standard [14]. The LOM standard was chosen for it's ease of use, extensiblity, and popularity in e-learning applications. Rules set in the semantic framework and pedagogical agents in the MAS dynamically update the learner profile with data about each student's learning state and behavior. The graph databases are then joined together to build semantic rules for personalisation and adaptivity. Web services are used to communicate to the LMS the personalisation of learning content and the reconfiguration of learner profiles.

*Personalisation on the WASPEC Platform*

Personalisation on the WASPEC platform is done in three stages and with a multiparameter approach. The phases determine what learners have to learn, how they wish to learn, and how well they have learned. The first phase is based on their level of knowledge (personal needs), the second on other preferences based on individual features, and the third on their performance and learning behaviour (personal development and individual characteristics).

Two indicators (LOR-PD and CRLO-PD) were designed to choose and combine personalisation parameters in order to enable multiple-parameter personalisation in a scalable

manner [15]. A personalisation strategy for a specific learning scenario is defined by the selection and combination of personalisation parameters. A personalisation strategy decides whether or not a learning object (the smallest unit of learning) is personalised. Figure 4 depicts the selection of various parameters to generate various learning strategies.

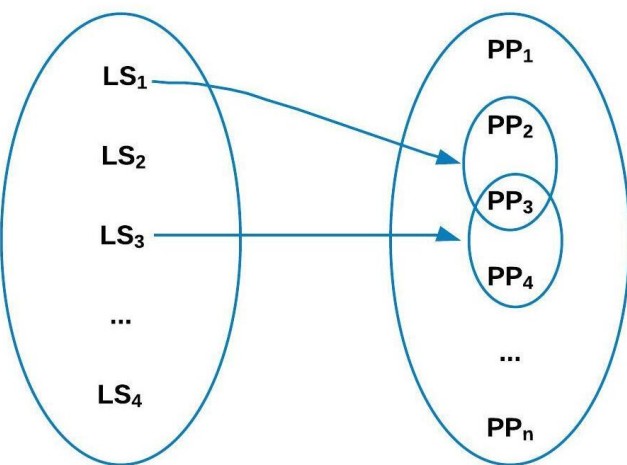

**Figure 4.** Selecting Relevant Parameters with the LOR-PD Index.

The LOR-PD (Learning Object Representation based on Dimensions of a Personalisation parameter) and CRLO-PD (Complementary Ratio of Learning Objects based on Personalisation parameter Dimensions for each competency) indices for a personalisation parameter demonstrate how a personalisation parameter is represented in a course depending on included dimensions. While the LOR-PD is important for identifying suitable personalisation parameters, as illustrated in Figure 4, the CRLO-PD defines which dimensions of a specified personalisation parameter are used for customisation of each learning object. Figure 5 depicts the combination process and the possible outcomes from applying the CRLO-PD function to each LO.

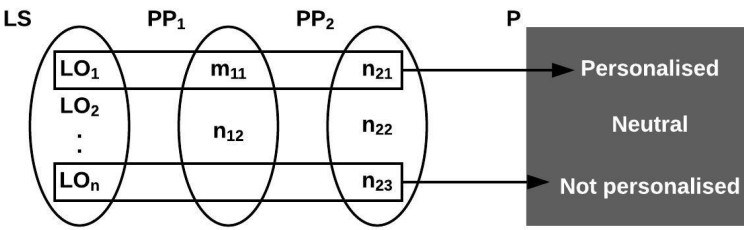

**Figure 5.** Combining Dimensions of Selected Parameters with the CRLO-PD Index.

The subject of "what the learner has to learn" is handled in the first phase of personalisation, which incorporates customisation based on personal needs, as learners are classed based on their degrees of knowledge. All learners have identical learning objectives, which is competency mastery, but each learner progresses at his or her own pace, which is determined by their degrees of knowledge. Higher knowledge-level learners can access learning objects below their levels, whilst lower knowledge-level learners can only access learning objects tagged with their levels.

The second phase of learning addresses the question of "how the learners learn" in order to achieve their learning objectives. This is accomplished via differentiated instruction, which generates multiple learning paths based on the characteristics given by the LOR-PD index [15]. The CRLO-PD index determines the personalisation of each learning object depending on selected parameters [15]. To personalise a course utilising a collection of available parameters, the LOR-PD index selects a subset that is relevant to the course, and the dimensions of the selected parameters are combined with respect to the CRLO-PD

index. This allows learners to learn in a variety of ways based on their preferences within a range of viable learning strategies [4,15].

As learners proceed through a course, the final phase of personalisation addresses the "how well learnt" question. The learning goal will be to gain sufficient mastery of identified competencies during the learning process. Students' profiles (knowledge level categorisations) are updated to reflect competency mastery as they proceed along their unique learning paths. Learners who are unable to achieve a specified degree of mastery for each knowledge level may have their preferences changed based on their interactions with other learning materials, enabling a different learning path toward the same learning goal.

## 4. Evaluating WASPEC

The first step in the evaluation process was to choose and personalise a course. An English Language course based on the Common European Framework of Reference for Languages (CEFR) curriculum was chosen for student experimentation. The abbreviated course was organised around four primary concepts and 33 curriculum competencies.

### 4.1. Personalisation on the Service Framework

The subsequent step was to add learning resources to Moodle in order to instruct the specified competencies. Learning resources are introduced to the Moodle platform in the same way that a regular course is. Through web-services on the Service Framework, metadata annotation from the LOM standard was added to these learning resources. This resulted in the addition of 64 learning objects to the course. The process of adding learning materials to the LMS and metadata annotation to course content on the SF is depicted in the Figure 6.

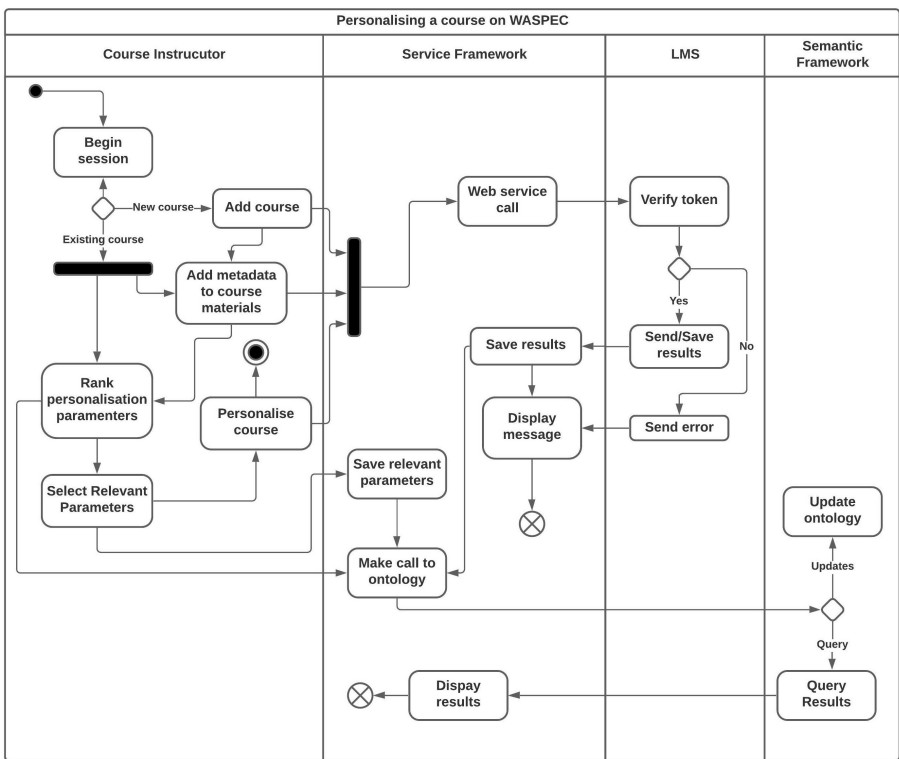

**Figure 6.** The Process of Personalising a Course on WASPEC.

The most relevant parameters can be ranked using the LOR-PD index based on an average time specified for each task on the Service Framework. Figure 7 depicts the ranking by satisfaction values using dynamic programming after adding time values to each parameter [15]. The Felder-Silverman Learning Style [16] categories of 'Visual/Verbal' and 'Active/Reflective' were chosen as suitable factors for personalisation for this course

based on these values. This is because all dimensions of both parameters are represented and have reasonable satisfaction values.

| Title | LOR-PD | Time |
|-------|--------|------|
| Media_Preference (2 of 4 dimensions) | 0.81 | 1 |
| Global_Sequential_FSLS (1 of 2 dimensions) | 1 | 3 |
| Visual_Verbal_FSLS (2 of 2 dimensions) | 0.81 | 3 |
| Active_Reflective_FSLS (2 of 2 dimensions) | 0.69 | 3 |
| Motivation_Level (1 of 3 dimensions) | 0.17 | 4 |
| Sensing_Intuitive_FSLS (0 of 2 dimensions) | 0 | 3 |

**Figure 7.** Personalisation parameters with LOR-PD indexes ranked with time (in minutes).

After the appropriate parameters have been selected, the SF is used to personalise each learning object on the LMS. The CRLO-PD index assesses if a LO item should be 'personalised', 'neutral', or 'not personalised' for both the Felder-Silverman learning style categories of 'active/reflective' and 'visual/verbal'.

Visualising the Personalised Course on Moodle

Figure 8 depicts the view of a course administrator on Moodle with different learning objects personalised based on knowledge level, which is the initial phase of customisation as indicated in Section 3.

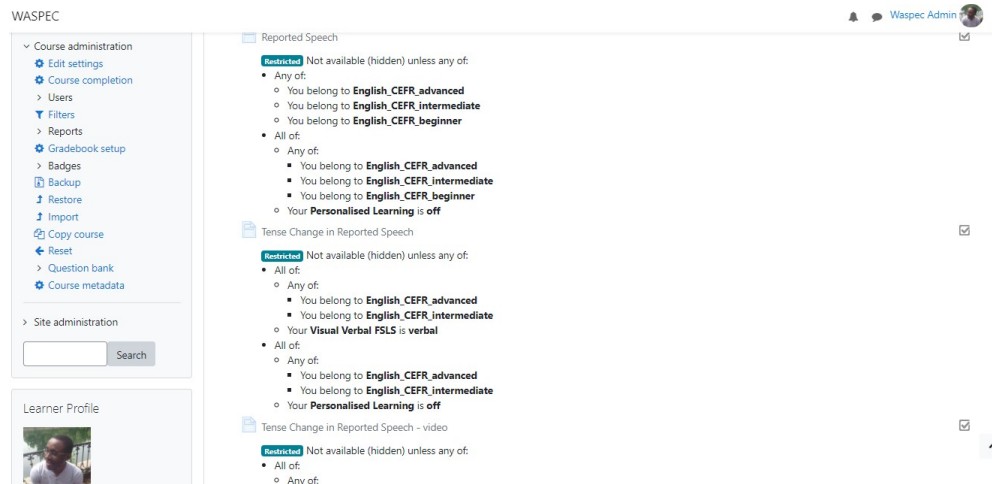

**Figure 8.** Phase 1 Personalisation on Moodle as viewed by a Course Administrator.

Figure 9 depicts the course instructor's perspective on two learning objects instructing the concepts "*Prepositions of time*" and "*Distinguishing between 'during', 'for', and 'during'*" at the second level of customisation.

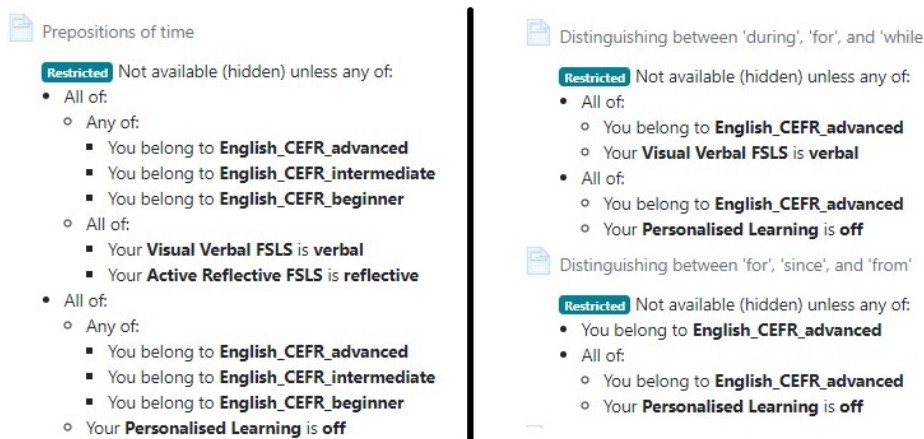

**Figure 9.** Phase 2 Personalisation on Moodle as viewed by a Course Administrator.

Figure 10 illustrates a learner's perspective with personalisation running in the background and concealed from the learner's view.

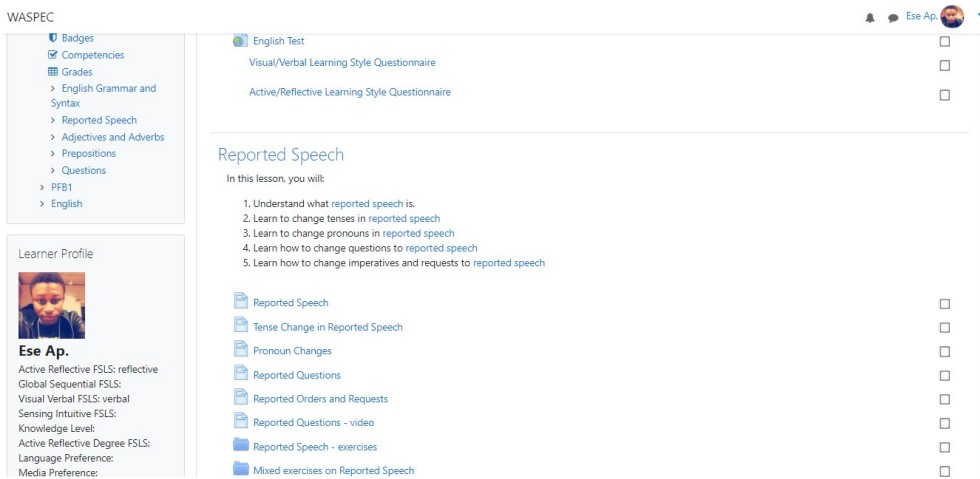

**Figure 10.** Course View of a Learner.

### 4.2. Demographics of Participants and Experiment Setup

The purpose of this experiment was to test the hypothesis, "*Learners can use the WASPEC learning platform, which offers personalised learning content based on their preferences, requirements, and interactions.*" Prior to its launch on 4 May 2021, the experiment was organised for approximately two months with the management of Petra Christian Academy, Ughelli, a secondary school in Delta State, Nigeria.

The initial phase in the experiment was to choose students who were willing to participate. The experiment was disclosed to the students throughout the recruiting process, and 24 individuals were initially chosen. However, several students withdrew before to the commencement of the experiment, while others were recruited to the group. Following the rearrangement, the experiment began with a total of 28 students, including 13 males and 15 females. Students were chosen from both the junior and senior sections, as shown in Figure 11, while the sole criterion for selection was that they had access to the platform through mobile devices or PCs.

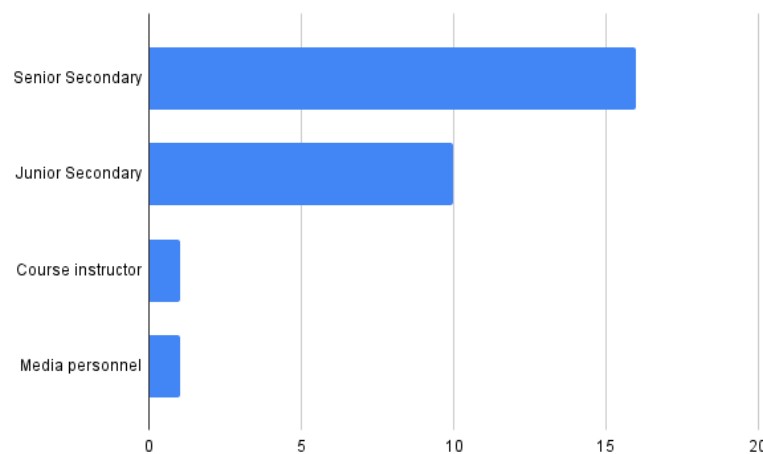

**Figure 11.** Demography of Participants.

Following the selection of students for the study, the following stage was to discuss the purpose of the study in detail using a 'Participant Information Sheet' (see Appendix A.1) and verbal explanations from research team members. Following that, all students signed a "Participant Consent Form" (see Appendix A.2) in order to receive ethical clearance; participants under the age of 18 had their parents or guardians sign the forms. Finally, 25 of the 28 students who started the experiment finished it. The experiment lasted roughly one month and two weeks and ended formally on 21 July 2021.

### 4.3. Experimental Procedure

The students began the experiment by completing a test to evaluate their knowledge levels based on the course plan. At the completion of the assessments, which took roughly 10 min for each student, there were no pupils in the beginner group, 11 in the intermediate group, and 17 in the advanced group.

Following the initial test to categorise learners based on knowledge levels, students answered surveys to establish their preferences depending on the course's relevant criteria for personalisation. With the initial knowledge test and preference activities completed, the learner profile was modified to reflect the learner's preferences and requirements. Following that, the student may explore customised learning content tailored to their interests and requirements. Intermediate students were to be promoted to the advanced group after they demonstrated mastery of the competencies, while advanced students were to complete the course when their performances demonstrated competency in the learning objectives.

Only two students (from the intermediate group) did not satisfy the requirements and had their learner profiles modified. They then improved their performance and were promoted to the advanced level, where they finished the course assessments as well.

### 4.4. Learner Experience Survey

The learners completed a survey based on their experiences with the system at the end of the course. The poll contained questions related to 'Usefulness', 'Attitude Toward Using', and 'Intention To Use'. Each question had alternatives corresponding to a five-point Likert scale: 'strongly agree', 'agree', 'neutral', 'disagree', 'strongly disagree'. Before taking the survey, the course instructor briefly discussed the questions with the students. There was also an open section to highlight issues and recommendations. Table 1 lists the questions that were included in the brief survey.

**Table 1.** Survey Questions.

| No. | Question |
|---|---|
| Q1 | The learning materials I received were appropriate for my level of knowledge. |
| Q2 | The learning materials I received were appropriate for my preferences. |
| Q3 | Learning on the WASPEC platform had a positive influence on my learning experience. |
| Q4 | I would encourage my school to adopt this form of personalised learning. |

*4.5. Internal Consistency and Acceptance*

Figure 12 depicts the learners' replies to all the survey questions, and Table 2 summarises the mean, median, and standard deviation for the learners' responses. The reliability of the test using $\alpha$ is also measured in the table.

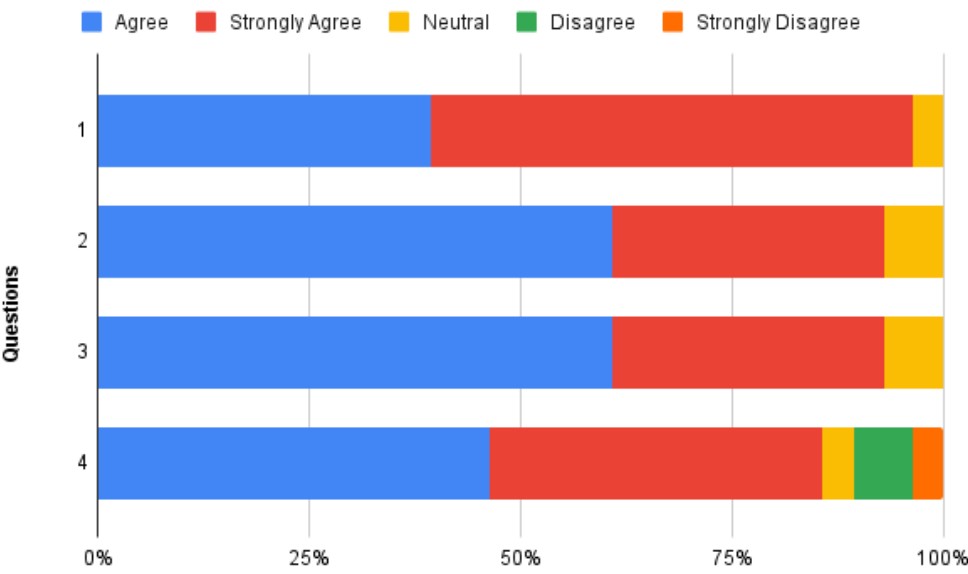

**Figure 12.** Results of the Survey Questions.

**Table 2.** Summary of the Learner Feedback.

| | Mean ($\bar{x}$) | Median | Standard Deviation ($\sigma$) | Cronbach's Alpha ($\alpha$) |
|---|---|---|---|---|
| **ATU** | 4.31 | 4 | 0.73 | 0.81 |

The experimental survey's variables were consistent, as evidenced by an $\alpha$ score of 0.81. The mean value was used to determine the learners' acceptance/satisfaction rate in terms of the platform's usefulness in terms of their level of knowledge and preferences, as well as their desire to embrace this kind of personalised learning. A mean of 4.31 and a median of 4 suggest that the great majority of learners were satisfied with their learning experience and welcomed this form of instruction.

*4.6. Learner's Comments*

Aside from the multiple-choice questions, students were encouraged to highlight any issues they found while learning or to provide recommendations. Some notable comments include:

*"It was a very nice experience and I hope to experience it again."*

*"I enjoyed it, but the network was somehow slow"*

*"My experience was nice, but a little stressful. It was also affecting my studies."*

*"I enjoyed the program. I am glad to be opportune to get such knowledge."*

*"I am so glad I joined this program. I was able to learn new things in English."*

*"I found it very useful."*

There were no significant technological problems reported because personalisation occurs totally in the background. The comment citing slow network issues was about an issue with the internet provider, not a technical problem with the platform. The student noted that the learning pages did not always load quickly. Another student found the activity stressful because it was not part of their regular curriculum.

## 5. The Architecture of WASPEC 2.0

As previously stated, integrating personalisation components into an existing open-source learning management system is an efficient approach of developing personalised adaptive learning systems. These systems frequently do not offer personalisation by default. However, despite the lack of features for customisation, they have very sophisticated administrative and learning functionalities that would be pretty difficult to recreate if a design were designed from the ground up. As a result, the WASPEC design supports modular components that can communicate with any LMS that supports web services. The Service Framework, Semantic Framework, and Multi-Agent System described in Section 2 are among these components that comprise the personalisation engine. Laravel was used to implement the Service Framework, which provides the personalisation of learning resources in the background of Moodle Learning Platform.

One of the comments made by participants in a prior review of the WASPEC platform [4] by researchers and instructors knowledgeable with personalised learning was that the process of personalisation on the platform might be too complex for a non-technical individual to work with. To overcome this issue, the next implementation and testing iteration recommends implementing the personalisation components as a plugin that can be deployed on Moodle. Moodle was chosen because it is a popular and widely used LMS with a variety of features that was designed with both pedagogy and technology in mind. Personalisation can be done directly on the Moodle platform using this architecture, rather than switching from one platform to another during the process. Because the architecture (Figure 13) is very modular, the personalisation components can be implemented with minor adjustments on a different LMS, such as ILIAS [17] (which was originally intended for this project).

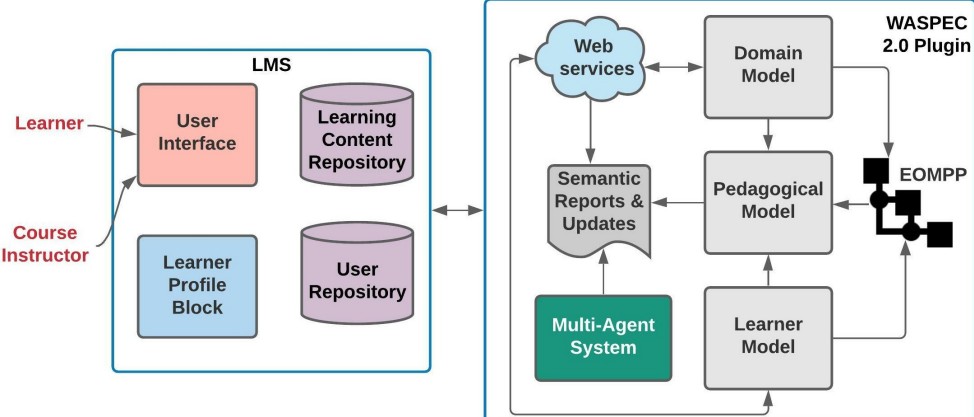

**Figure 13.** The Architecture of WASPEC 2.0.

The architecture incorporates similar components that were part of the initial WASPEC implementation's framework [4]. The primary distinction is that the SF is no longer included in this design. The learner model stores critical information about a learner on

the learning platform at all times. The domain describes the relationship between courses, competencies, and LOs. The pedagogical model specifies guidelines that enable rule-based reasoning for personalisation and adaptivity of learning content.

EOMPP describes a slightly more improved version of the set of ontologies used for the design in the first implementation [4]. It describes the links between the different models: domain, learner, and pedagogical. The element representation in EOMPP enables knowledge modelling, semantic annotation of learning resources, and semantic reasoning. EOMPP is made up of various ontologies, as illustrated in Figure 14.

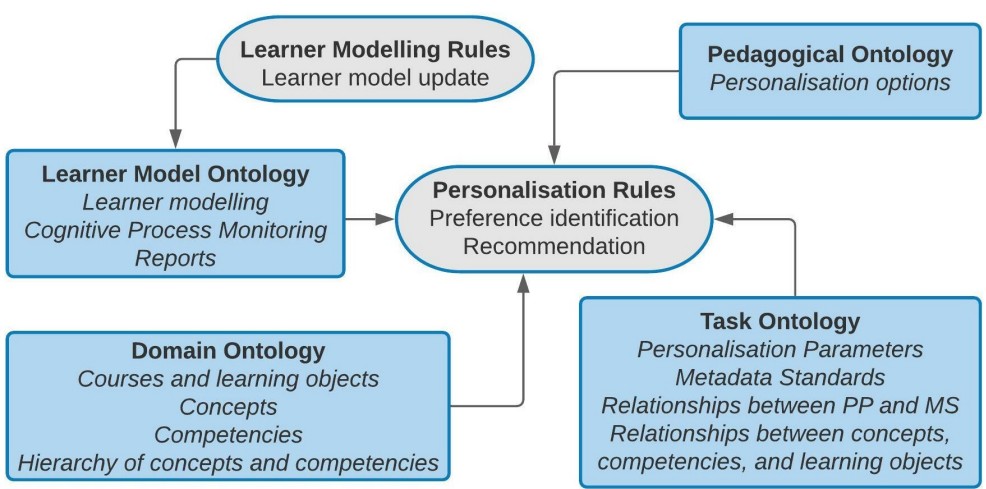

**Figure 14.** Ontologies and Rules in WASPEC 2.0.

The learner and domain ontologies are created by using the D2RQ platform [18] to map the learner and domain repositories as RDF schemas. The learner model ontology holds generic and adaptive learner information. The domain ontology is defined as a taxonomy of concepts and competencies, with features and interconnections that connect them to other concepts in a hierarchical order. The link between personalisation parameters and metadata components is described by the task ontology. It also illustrates how user features connect to domain model elements. Finally, there is the pedagogical ontology, which provides personalisation possibilities, as well as guidelines to follow when making adaptive judgments.

The agents in the MAS are responsible for tracking learner progress, updating learner profiles, providing adaptivity, and updating the domain. These updates are made possible in EOMPP by OWLready2 [19], a Python module that provides access to OWL ontologies.

## 6. Discussion and Related Work

There have been several attempts in the literature to create educational systems to enable personalised and adaptable learning, with researchers taking various techniques, as stated in previous sections. The approach employed in the creation of the WASPEC platform is unique in that various tools commonly used in educational systems are combined. According to a review of current trends in personalised adaptive educational systems by Somyürek [20], modular frameworks, semantic web technologies, and standardisation are some of the most frequent technologies utilised to facilitate interoperability and scalability in the development of such systems.

Modular frameworks are integrated into the WASPEC platform through the usage of Moodle LMS and pedagogical agents. Modular frameworks "permit individuality, flexibility, exchangeability, and the extension of a system", and "they are seen as an effective solution for shortcomings found in complete learning designs" ([21], as cited in [20]). In the literature on e-learning systems, the semantic web has become an increasingly attractive research area. The technologies are divided into two sub-themes: ontologies and query language. Semantic web technologies enable the formalisation and sharing of knowledge

across several platforms. EOMPP and SWRL rules that describe personalisation choices and relationships between different elements on the platform are used to incorporate semantic web technologies. Finally, standardisation aims to create "a set of widely adopted and interchangeable features, to make instructional contexts compatible with each other, interoperable, and repeatable" [20]. The LOM standard elements essentially specify how the domain is configured on the WASPEC platform.

Some platforms in the literature are designed exclusively with a multi-agent system network using modular frameworks. MASHA-EL (Multi-Agent System Handling Adaptivity for E-learning), in which learners use a variety of devices, is assisted by a device and teaching agent [22], a ubiquitous teaching assistant (u-TA), capable of assisting students in problem-solving activities during laboratory sessions [23], and AILS (Adaptive Intelligent Learning System), developed with JADE (Java Agent Development Framework), which provides content, presentation, participation, and perspective adaptation [24], are examples of such designs.

Personalisation via extending LMSs includes designs such as MAL (Moodle Adaptive Learning), which allows learners to be customised based on their learning styles [25]. MAL was developed as an ontology to define the connections between learners, learning resources, and learner actions on the platform. SWRL rules were also developed to identify adaptive functions during learning. Nafea et al. [26]'s AAST (Arab Academy for Science and Technology and Maritime Transport) Moodle implementation presents a personalisation model based on the Felder-Silverman Learning Style model and the Myers-Briggs personality type indicator. Personalized Learning Management System (PLeMSys), a design by Blazheska-Tabakovska et al. [27], allows learners to obtain customised learning experiences on Moodle via plugins based on their degree of competence and learning style. This implementation specifies two levels of personalisation: the first is based on the student's knowledge level and the second on their learning methods.

There is the popular Protus (PROgramming TUtoring System) implementation with semantic web technologies, a Java programming language teaching system that recommends effective learning resources to learners depending on their learning preferences [28]. TANGRAM, an intelligent information system, is yet another ontology-based design. Learning resources are customised to the learner's current knowledge level, learning style, and personal preferences [29]. Finally, Rule-PAdel (Rule-based personalised adaptive e-learning system) is a TANGRAM-inspired architecture that dynamically generates personalised learning content from reusable learning content using a semantic rule-based approach [30].

In the design of the WASPEC platform, one of the main research goals was to accommodate multi-parameter personalisation in an effective and scalable manner. This necessitated the definition of the indexes LOR-PD and CRLO-PD to integrate multiple parameters in any domain. Table 3 compares the WASPEC platform to other similar designs in the literature.

**Table 3.** Comparing the WASPEC Platform with Similar Designs in the Literature.

| Platform | Modular Framework | Semantic Web Technologies | Standardisation |
|---|---|---|---|
| WASPEC | Pedagogical agents and LMS (Moodle) | Ontologies and SWRL rules | IEEE LOM Standard |
| AAST Moodle | LMS (Moodle) | Ontologies and SWRL rules | - |
| MAL | LMS (Moodle) | Ontologies and SWRL rules | Dublin Core, IEEE Lom Standard |
| Protus | - | Ontologies and SWRL rules | - |
| TANGRAM | - | Ontologies | IEEE LOM Standard |
| Rule-PAdel | Instructional Objects | Ontologies and SWRL rules | - |

### 7. Conclusions and Future Directions

This study presents an evaluation of WASPEC, a Moodle-based modular platform that delivers personalised learning content to learners using a multi-parameter personalisation technique. WASPEC employs two metrics (LOR-PD and CRLO-PD) for selecting and combining parameters for personalisation based on their significance in each course. The LOM standard is used to annotate learning resources, while the EOMPP ontology is used to represent relationships between distinct elements of personalisation in the architecture. The evaluation results reveal a good level of acceptance and consistency with the use of the TAM questionnaire.

The current implementation described accomplishes personalisation of learning resources using an external platform that uses web services and can interoperate with LMSs other than Moodle. The annotation of learning resources on an external platform, which serves as the adaptive engine, was time-consuming and burdensome for non-technical course instructors during the first iteration of implementation and evaluation. As a result, at the end of this paper, a novel implementation technique is proposed that incorporates all features and components of personalisation in a plugin that can be installed on Moodle. While this retains the modular architecture and the concept of separating the business logic of personalisation from the basic functionalities of an e-learning platform, it gives a simpler approach to personalisation for course instructors.

The current evaluation with learners was performed with only 25 students. Future work would include implementing the recommended design and conducting additional testing with more learners in order to give a more comprehensive examination of the acceptability and utility of personalised learning resources in learning scenarios of the WASPEC platform.

**Author Contributions:** Conceptualisation, software, resources, data curation, methodology, writing—original draft preparation, investigation, U.C.A.; writing—review and editing, supervision, G.C.C. All authors have read and agreed to the published version of the manuscript.

**Funding:** This research received no external funding.

**Institutional Review Board Statement:** Not applicable.

**Informed Consent Statement:** Informed consent was obtained from all subjects involved in the study. See Appendix A.1.

**Data Availability Statement:** The form used in this study can be accessed with this link: https://forms.gle/czd61pjyVfvjwRs27 (accessed on 5 June 2022). The responses for this study can be available here: https://docs.google.com/spreadsheets/d/1otoYVT2ZLgSmyuGIl169ekjXfZuWWjpebITKe9k1zB0/edit?usp=sharing (accessed on 5 June 2022).

**Conflicts of Interest:** The authors declare no conflict of interest.

### Abbreviations

The following abbreviations are used in this manuscript:

| | |
|---|---|
| EOMPP | E-learning Ontology for Multi-Parameter Personalisation |
| ICT | Information and Communications Technology |
| LO | Learning object |
| LOM | Learning Object Metadata |
| LMS | Learning Management System |
| MAS | Multi-Agent system |
| MOODLE | Modular Object-Oriented Dynamic Learning Environment |
| WASPEC | Weighted Agent System for Personalised Curriculum |

**Appendix A**

*Appendix A.1. Participant Information Sheet*

*Please read this information sheet carefully to understand the details of this study.*

**Project Title:** Personalising an online course with multiple parameters

**Purpose of the research study:** The purpose of this research study is to test an e-learning platform (WASPEC) that provides personalised learning content to students based on their preferences and abilities.

**Lead researcher:** Ufuoma Chima Apoki is a postgraduate researcher at the Faculty of Computer Science, Alexandru Ioan Cuza University, Iași, collecting data as part of his doctoral thesis.

**What you will do in the study:** Participants will be given learning materials for an English language course based on the Common European Framework of Languages (CEFR). Before beginning the course, participants will be required to complete tests and questionnaires to determine their abilities and preferences. Then, based on the results of the tests and questionnaires, participants will go through the learning content, which will be personalised. After a period of studying the materials (determined by the course instructor), tests will be administered to determine participants' knowledge level of the course concepts.

**Time required:** The study will be held during your school session but will not interfere with your regular academic or extracurricular activities. It will take approximately three to four months.

**Risks and Benefits:** There are no anticipated physical, psychological, academic, or personal risks or hazards associated with taking part in this study. However, if you believe you are at risk, you have the right to withdraw from the study at any time. However, there are educational benefits because this research and experience will expose you to novel e-learning techniques and paradigms.

**Confidentiality and Participation:** The data you submit will be treated discreetly in this study. The information you provide will be anonymous, which means that your name will not be associated with the information. Even if someone tries, it will be impossible to deduce your identity from the data. Your data, feedback, grades, and responses will be reported in a way that does not reveal your identity. Your participation in the study is entirely voluntary, and you have the right to withdraw at any time without prejudice or penalty.

**How to withdraw from the study:** Any participant shall be able to withdraw by notifying the principal researcher or the course instructors.

**Appreciation:** For participation in the study, there will be no monetary advantages or anything similar. Your participation in the progress and completion of this research is, nonetheless, highly welcomed and valuable.

*Appendix A.2. Participant Consent Form*

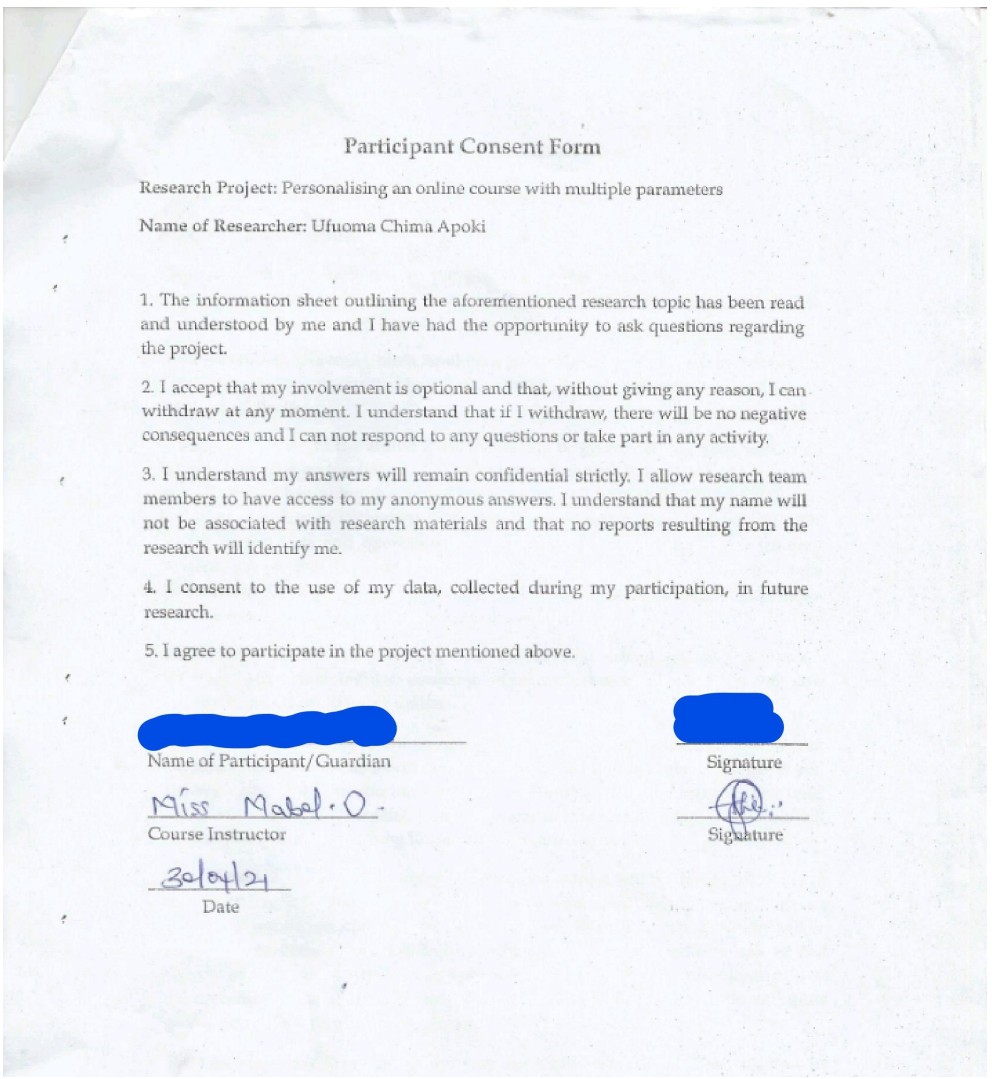

**Figure A1.** Sample of the Consent Form.

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
