# Peer review of "A Modular and Semantic Approach to Personalised Adaptive Learning: WASPEC 2.0"

_applsci, doi:10.3390/app12157690_

Round 1

Reviewer 1 Report

The intro is a bit abstract, a very simple and short example could be given. How is the business logic "entangled with the systems' basic operations?"

The questions that the pupils were asked seem to be a bit complex for them. For example if the content is appropriate. Does that mean if they understood everything, something, how much? How do you differentiate, are there any degrees except for strongly disagree vs disagree vs agree vs strongly agree?

To me it appeared that the article finishes abruptly. we do not get top know how each pupil's experience was customised - or perhaps this stage has not yet been achieved?

I still do not understand how this idea, which seems very interesting, actually works in practice. Are there many alternatives of the material of a course available? Are there extra exercises that can be provided to students that 'strongly disagree' that the material is 'appropriate' to them?

to me this is unfinished work, perhaps the scope and contribution  should be made clear in the article.

All in all, an interesting and potentially valuable research however it is unclear to me how it works in practice, in a very concrete and complete example

References seemed scarce and somewhat self-citing (the latter may be ok in fact to provide background but reader should no be forced to go there to get all the info required to understand this article)

Author Response

  • The introduction has been improved to include examples and provide more explanations on the ambiguity of the business logic being "entangled with the systems' basic operations.
  • More details have been added on the section on the experiments. The questions the students took in the survey were explained to them before the exercise.
  • Section 4 has been improved with an additional subsection to describe and illustrate how personalisation works and the views from the course administrator and learners.

Reviewer 2 Report

The experimental research holds promising prospect to impact positively on learning through mobile and computer devices and the content served on WASPEC. A greater level personalized learning content based on learners' preferences, requirements, and interactions is offered through WASPEC 2.0 experiment presented in this study.

The authors, however, has demonstrated a sound understanding of the field and object of research. The author can improve this paper if the included some more participant information in the result section, for example, what were the actual responses from more participants. The authors have only stated comment from one participant as a way of feedback of the WASPEC 2.0. Although this paper is quantitative, a blend of relevant qualitative data description in the result section will pull in the richness of the participants experiences and the contributions that study intends to make to personalized learning experience.

Author Response

  • Section 4 has been improved to include more participant information and the personalisation process.

Reviewer 3 Report

accepted in this version

Author Response

The paper has been revised to be more comprehensive and less ambiguous.

Reviewer 4 Report

This paper addresses the topic of personalised adaptive learning. In particular, the authors evaluate an existing approach (WASPEC system) and propose an extension called WASPEC 2.0. While the topic is interesting and worth investigating, there are several important drawbacks of the paper:

1. There is no comparison with previous approaches for personalised learning.

2. More details about the experiment are needed. For example, which subject were students learning? And how learning "conflicted with other academic pursuits"?

3. The evaluation of WASPEC 1.0 was positive, users were happy with it. Therefore, why is a new version needed? A convincing motivation is needed.

4. The sample of the empirical study is very small, only 25 students, making the results not very significant.

5. The parts of the system are not evaluated separately. An ablation study of the system, studying which parts of the system are actually helpful, would be very interesting. For example, do semantic annotations actually provide a difference in the evaluation of the adaptive learning?

6. WASPEC 2.0 is described very vaguely. For example, 4 ontologies (learner, domain, task, and pedagogical) are mentioned but it is not clear who should build them, the expected size of each of them, the cost of building them, the reusability, etc. For example, if the ontology schema is always the same, and only the data corresponding to each student change, this should be clarified. How ontologies are updated should also be clarified. Some examples of learner modelling rules and personalisation rules would also be very helpful. Currently, their purposes and the language/s used to represent them are unclear.

7. WASPEC 2.0. is not implementation and evaluated.

I could accept letting points 4, 5, and 7 as future work, as they involve a significant amount of work. However, points 1, 2, 3, and 6 must be addressed.

Author Response

  1. Section 6 has been included to discuss similar designs and compare them with the WASPEC platform.
  2. More details have been included in Section 4 to describe the experiment and the participants' experiences.
  3. Section 5 has been improved to include a more comprehensive motivation for a newer version of the platform.
  4. Future work includes testing with a larger sample of learners.
  5. Section 5 includes reference to the initial evaluation of the system by course instructors and researchers.
  6. The description of the architecture of WASPEC 2.0 has been improved in Section 5 to a more comprehensive plan of the proposal.
  7. Implementation and evaluation of WASPEC 2.0 is part of future work

Round 2

Reviewer 4 Report

This paper addresses the topic of personalised adaptive learning. In particular, the authors evaluate an existing approach (WASPEC system) and propose an extension called WASPEC 2.0. This revised version has clearly improved the paper and can be accepted.